# Identity Management in Future Railway Mobile Communication System

**Evelina Pencheva** [1,*]**, Ivaylo Atanasov** [2] **and Ventsislav Trifonov** [2]

1 Telecommunications Department, Faculty of Telecommunications and Electrical Equipment in Transport, "Todor Kableshkov" University of Transport, 1574 Sofia, Bulgaria

2 Communication Networks Department, Faculty of Telecommunications, Technical University of Sofia, 1756 Sofia, Bulgaria; iia@tu-sofia.bg (I.A.); vgt@tu-sofia.bg (V.T.)

* Correspondence: evelina.nik.pencheva@gmail.com

**Abstract:** The Future Railway Mobile Communication System (FRMCS) has emerged as a worldwide standard for railway communication. This technology enables the operational efficiency and safety of railways to be improved by providing mission critical communications, machine-type communication for the railway system on board, in addition to trackside telemetry and broadband connectivity for passengers. Different equipment types, users, and functional identities can be involved in communication, and each of them is uniquely identified. Identity management is an important part of the security functions provided by the FRMCS system. This paper presents a service-oriented approach to identity management functionality, enabling service composition for railway applications and service virtualization. This paper studies functionality for the initial registration and subsequent deregistration of railway devices, users, and their functional identities, in addition to the transfer of the registered identities between different FRMCS serving areas while the train moves. Two FRMCS services that follow the principles of representational state transfer architecture are proposed. Services' functionality is illustrated by use cases, data types, and application programming interfaces that enable services to be interacted with. Identity registration status models are developed, formally described, and mathematically verified. Discussion of the applicability of the proposed services for the implementation of FRMCS security and safety functions is provided. The presented service-oriented approach features a satisfactory level of flexibility and versatility.

**Keywords:** railway equipment; user; functional identity; registration management; service-oriented architecture; application programming interfaces; state machines

## 1. Introduction

The future of the rail transport sector depends on reliable mobile broadband connectivity, powered by innovative fifth generation (5G) network technologies for more ecological, safer, and faster travel. The railway industry has adopted advanced 5G technologies to improve overall service operation by improving efficiency and passenger experience, generating higher train speeds by enabling autonomous trains, and increasing passengers and staff security by making the systems more intelligent [1,2]. These connectivity improvements enable critical railway applications for automatic train control and protection, train localization, onboard system monitoring, and trackside equipment maintenance. The increased security provides access to real-time monitoring to ensure passenger safety and to protect assets. The advances in automation and data analytics for increased safety and efficiency stimulate the introduction of new applications.

Critical railway operations are vital to system operation, and by adopting 3GPP standardization for mission critical communications, both railways and passengers may benefit from standardized building blocks and devices. The Third Generation Partnership Project (3GPP) standards for mission critical communications define the architecture and

open interfaces that enable a high level of interoperability and simplify interconnectivity with future broadband networks [3,4]. Mission critical voice, data, messaging, and video services are key elements for implementing the vision that rail systems have for their future mobile communication systems.

Railway communications require high network availability, guaranteed quality of service for priority traffic, improved passenger safety through always-on communications, and reduced operating and maintenance costs. Technologies that support 5G connectivity and design principles such as network virtualization, network slicing, network softwarization, and mobile edge computing can address the specific requirements of railway networks [5–8].

The Future Railway Mobile Communication System (FRMCS), standardized by the European Telecommunications Standards Institute (ETSI) and the International Union of Railways (UIC), forms the basis of next-generation intelligent railways. It provides the means to improve rail operations and to offer innovative services to users. FRMCS standards define user requirements based on analyzing typical railway use cases [9]. Each use case is defined as a list of actions describing the interactions between the users and the system to achieve a goal [10]. The use cases related to critical voice communications include a feature called Role Management and Presence as a part of the user authentication and authorization process. The FRMCS standards also define the system architecture by analyzing the architectural requirements and the requirements for identification and addressing system security, and positioning [11]. The FRMCS system architecture creates clear separation between the application stratum, service stratum, and underlaying transport stratum. The research presented in this paper is positioned in the service stratum.

The FRMCS standards do not postulate how the identified railway requirements might be implemented. Future FRMCS standardization activities will include defining the building blocks and interfaces, provisioning communication services to the application layer, and ensuring interoperability. In this paper, a service-oriented approach for the implementation of the Role Management and Presence feature is presented. Two services for railway identity management are depicted for use as service bricks in order to realize FRMCS security mechanisms.

Existing works focus on problems related to railway security risks [12,13]. Exposure to potential malevolence can cause substantial impacts on human safety and infrastructure. The potential threats, vulnerabilities, challenges, consequences, and risk management, related to cybersecurity in railway systems, and the research methods, that may be used to achieve data security, are discussed in [14]. The security requirements for railway operations in [15] are identified based on a use case analysis that includes level-crossing, trackside, and rolling stock monitoring. A reference security architecture is also proposed that focuses on measurement data flows from sensors towards a data center for further data analysis and predictive maintenance. An analysis of the security threats and vulnerabilities of train control and monitoring systems and video surveillance systems is provided in [16]. The authors grade the risk levels according to their impact and consequences on the whole system.

Dealing with malevolent and accidental risks, that may compromise human safety and the functioning of trains, requires resilience-based security approaches that include authentication, authorization, integrity, and confidentiality. The existing identity authentication mechanisms in communication-based train control adopt a centralized key management system that is susceptible to single-point failures. In [17], the authors propose the deployment of a blockchain-empowered distributed security scheme to improve system security. In [18], the authors study a trust model, attack models, and security objectives focused on securing the communication between the train driver and the control center. An identity-based signcryption scheme and an identity-based signature scheme are proposed to achieve authentication, confidentiality, integrity, and end-to-end security. To ensure the security of the service information released at metro stations, a scheme to audit identity based on face

recognition is proposed, which improves the reliability of information sources and avoids potential risks caused by the display of illegal information [19].

The contribution of the proposed service-oriented approach for identity management is to present orchestrated usage of the existing security mechanisms in the FRMCS service stratum. The orchestration takes the form of signaling procedures that use message exchange in order to achieve the required security level. The functionality of identity management is described as services. The advantages of the proposed service-oriented approach are as follow:

- Cost savings, as cloud delivered services may be used by application developers without any specific knowledge of railway operation. The proposed services offer basic capabilities and may be used as building blocks in different railway applications.
- Capability to use the latest security tools and resources, as they may be embedded in the service logic while the interfaces and information model stay intact.
- Faster provisioning and greater agility because as-a-service solutions provide access to security tools and may be scaled up or down as required when there is excessive volume of traffic.

The rest of the paper is organized as follows: The next section describes basic terms that are used in the use case definitions and focuses on the required functionality of role management and presence as defined in [10]. Section 3 presents a service-oriented approach for the initial registration of a device, user, and functional identities. A new FRMCS service is proposed that enables applications deployed in FRMCS equipment to login/logout of the FRMCS system and to register/deregister their functional identities. The service also enables users to login/logout of the FRMCS system and to register/deregister one or more user functional identities. Section 4 presents aspects related to service implementation. Section 5 studies issues about the equipment and user mobility between different FRMCS systems and the necessity to relocate their registered identities. The section also presents a new FRMCS service that enables registered identities to be transferred between FRMCS systems as trains move across different FRMCS serving areas. The applicability of the services when implementing FRMCS security and safety functions is discussed in Section 6. The Conclusion summarizes the contributions and highlights the benefits and limitations of the proposed approach while outlining directions for future work.

## 2. Identities and Registration Management in FRMCS

According to [10], a functional identity describes a function performed by a user who is involved in a communication. It is used to address a user/system by identity or function instead of by user subscription or FRMCS equipment identity. FRMCS user equipment (UE) is a device that can use the FRMCS. There are different types of user devices that can be used in rail systems. The following are types of UE [10]:

- UE without a man–machine interface that does not require a functional identity, such as a "Sensor",
- UE without a man–machine interface that requires a functional entity, such as "Public Address System (PA system)",
- UE with a man–machine interface that is used in a train by a train driver, such as a "Cab radio",
- UE with man-machine interface for general purposes such as a "Handheld".

When UE is switched on, it attaches to the FRMCS network at the telecommunication level. The UE is reachable via its subscriber identity address. The associated FRMCS applications start up and are ready for use.

The UE registers itself in the FRMCS system at the application level, and the FRMCS identifies the type of UE. This is the first step of registration for UEs of all types. "Sensor"-type UE is reachable via its UE identity and subscriber identity address, and no other actions are required, as shown in Figure 1.

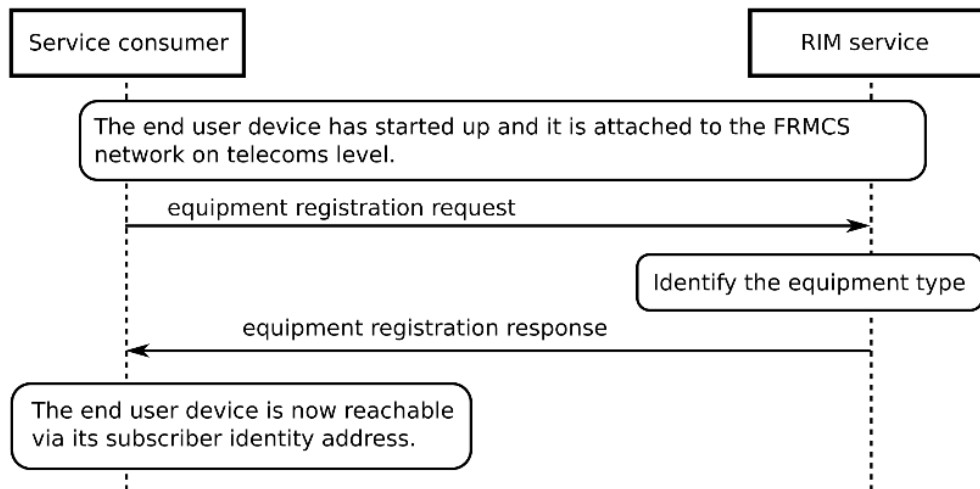

**Figure 1.** Flow of the registration for a "Sensor"-type FRMCS UE.

The second step of registration is for FRMCS UEs of "PA system" and "Cab radio" types. This step includes registration of UE functional number(s). The UE functional number is related to the equipment and points where the device is used, e.g., a PA system on a train.

The role management process of "PA system"-UE is shown in Figure 2. A UE may have more than one functional identity. The UE can be addressed via its functional identity and via the identity address of its subscriber.

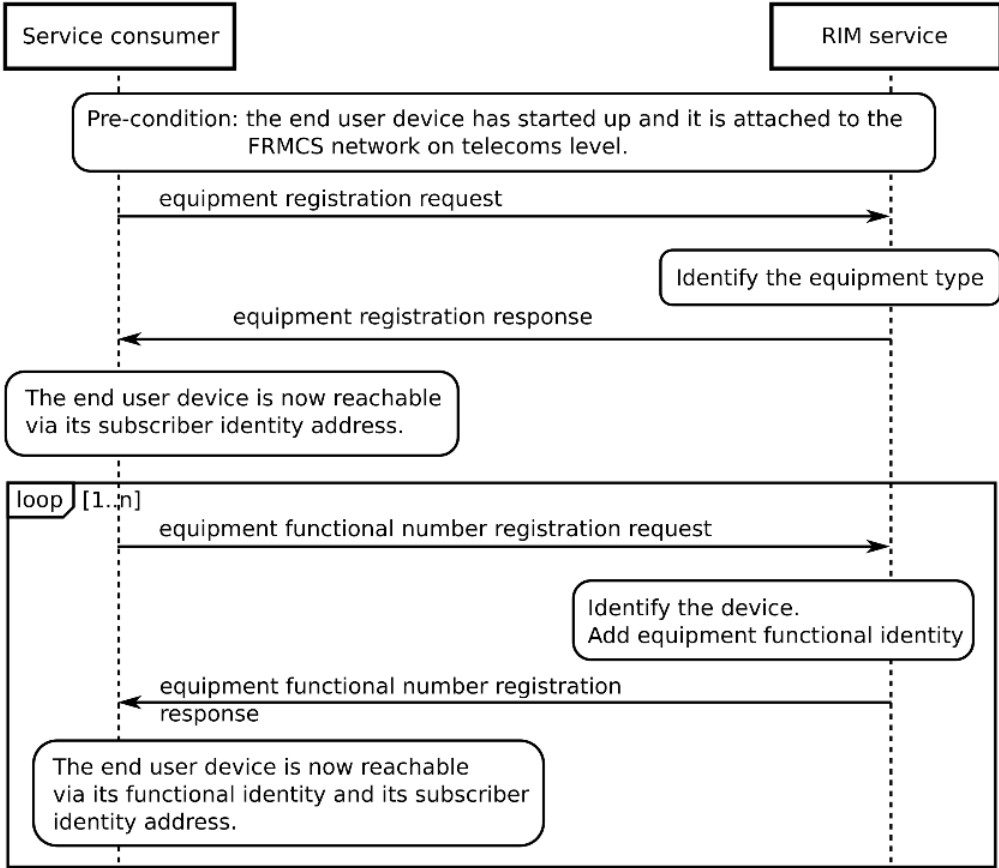

**Figure 2.** Flow of the registration for a "PA system"-type FRMCS UE.

Next steps of registration are for UE of "Handheld" and "Cab radio" types. Third step of registration includes user login in to the FRMCS system. The user logs into the FRMCS system using a man–machine interface. The user logs via his/her credentials, which can be functional or personal. The user login procedure may require a username and password, smart card, fingerprint, or eye scan depending on the UE's capabilities. The fourth step of registration includes registering of user functional number(s). This functional number is related to the user and identifies his/her functional role (e.g., controller, train driver, train staff, etc.). A user may have more than one functional identity.

Figure 3 shows the registration of "Handheld"-UE and Figure 4 shows the registration of "Cab radio"-UE. "Cab radio"-UE is reachable via its UE identity, its functional number(s), user identity, user functional identity(ies), and the identity address of its subscriber.

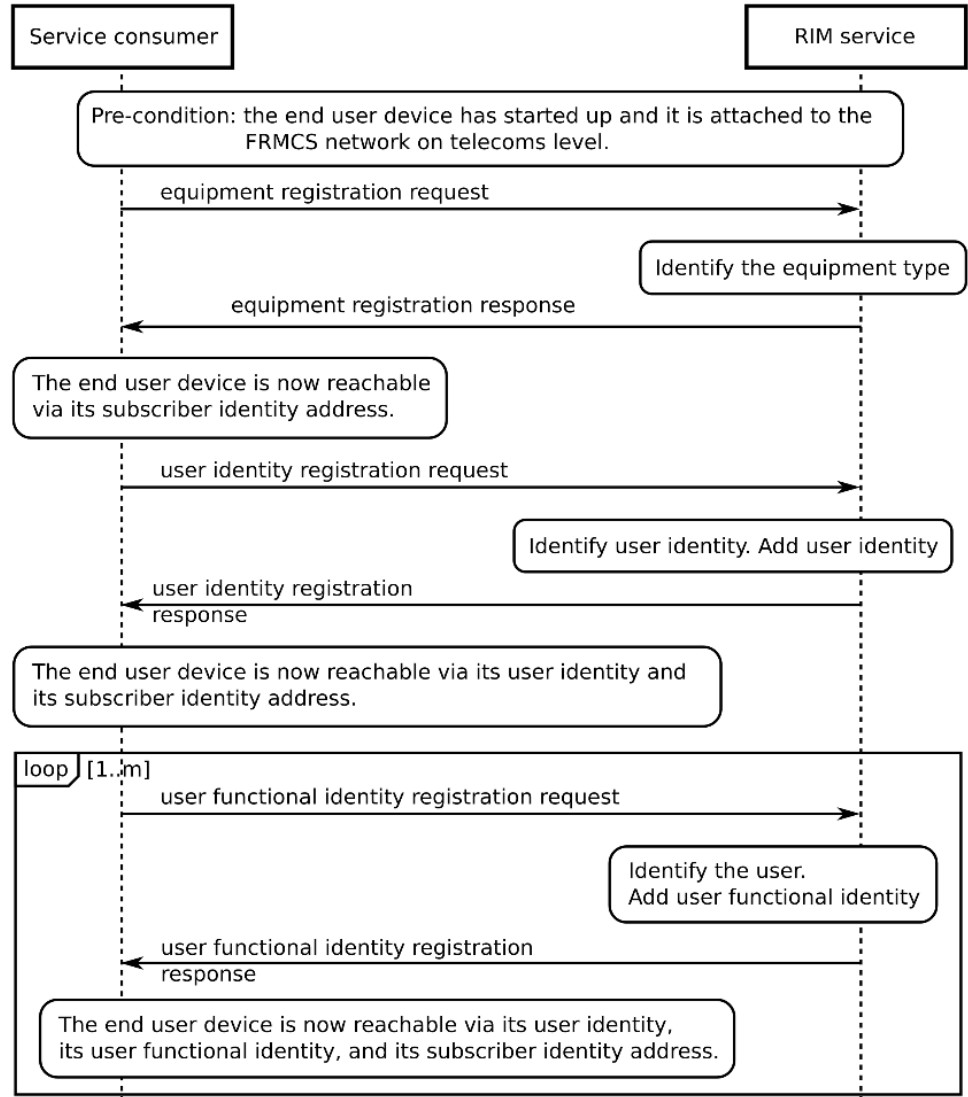

**Figure 3.** Flow of registration for a "Handheld" FRMCS UE.

In the FRMCS system, a user or equipment can deregister one or multiple functional identities. The deregistration procedure must be performed to change a functional identity and in this case the user deregistration can be driven by the user.

A user or UE can be identified by others who are involved in a communication. The available identities of the communication initiator are sent to the UE(s) of the so-called party(ies), and the receiving UE presents the identities based on a context (e.g., destination, time, status, location, etc.). A user or UE can be identified by others within a specific context (e.g., train, all drivers at a station, railway emergency communications, etc.).

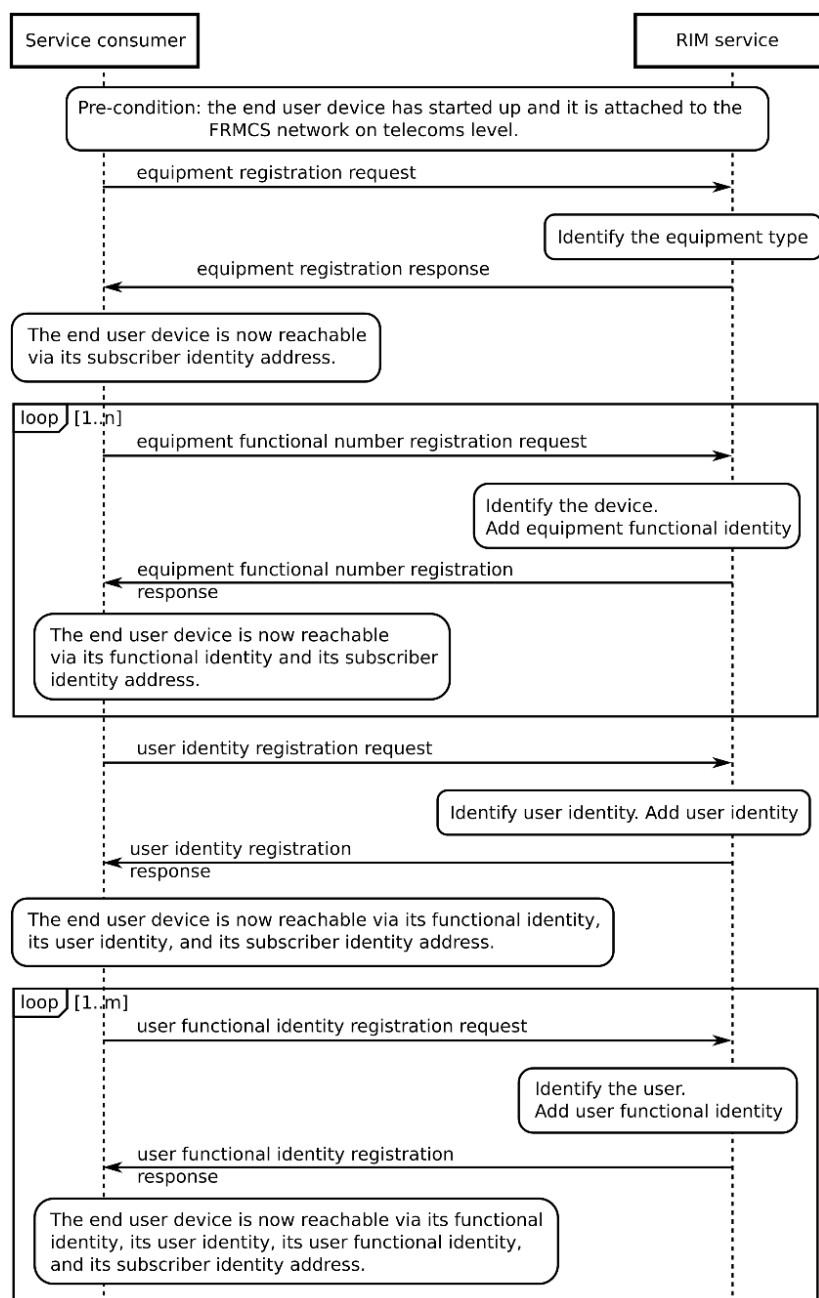

**Figure 4.** Flow of registration for a "Cab radio" FRMCS UE.

## 3. Description of Railway Identity Management Service

The proposed Railway Identity Management (RIM) service can be used for the following:

- The registration and deregistration of FRMCS UE;
- The registration and deregistration of an FRMCS UE's functional identity;
- The registration and deregistration of an FRMCS user's identity;
- The registration and deregistration of an FRMCS user's functional identity;
- The retrieval of registered UE identity(ies), UE functional identity(ies), user identity(ies), and user functional identity(ies);
- Subscriptions to notifications about changes in the registration status of UE identity(ies), UE functional identity(ies), user identity(ies), and user functional identity(ies).

The service's application programming interfaces (API) are bound by the Hypertext Transfer Protocol (HTTP) and are designed using the REpresentational State Transfer (REST) architectural style.

Figure 5 shows the structure of the RIM service resource Uniform Resource Identifiers (URIs), where the RIM service is registered in a service directory.

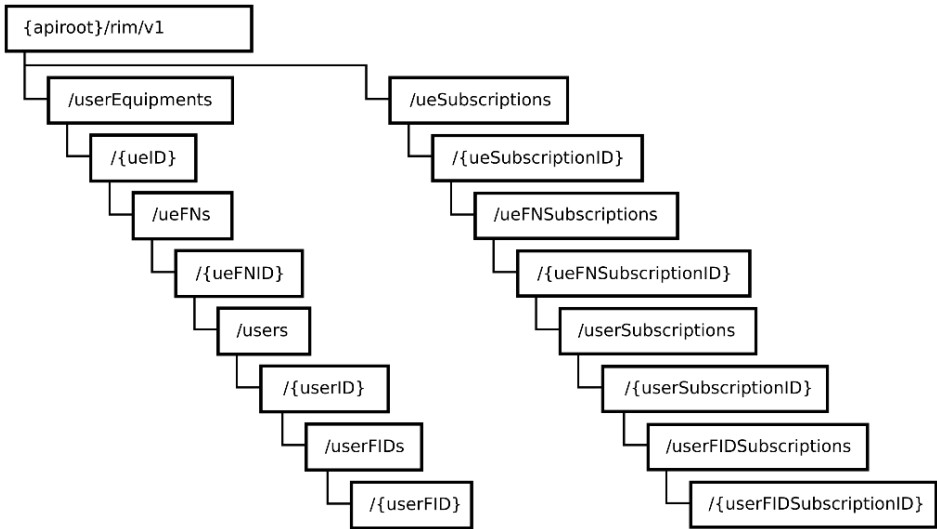

**Figure 5.** The structure of the RIM service resource URIs.

The userEquipments resource represents a list of all of the registered UEs. An HTTP GET method retrieves the list of registered UE identities, and an HTTP POST method creates a new ueID resource. The userEquipments resource is represented by a ueList data structure, which is a structured collection of UE information (ueInfo) lists and a self-referring Uniform Resource Locator (URL).

The ueID resource refers to individually registered UE information and it is represented by ueInfo, which is a structure comprising the UE address, UE type, UE presence status, and a self-referring URL. The UE presence status shows the UE context status in terms of activity (available, busy, railway emergency communication, etc.), placeType (train, station, office, etc.), placeProperties (the place where the user is currently), timeOffset (the number of minutes from the Coordinated Universal Time—UTC), location (direction, destination, track section), activeApplications (describes the active applications), etc. The presence information is provided by the FRMCS system. An HTTP GET method retrieves information about the individual UE, and an HTTP DELETE method deletes individual UE registration.

Figure 6 shows a scenario in which a new ueID resource is created, where information about individually registered UE is retrieved, and where UE is deregistered using an application. To register a new UE, an application creates a new ueID resource under http://\{apiRoot\}/rim/v1/userEquipments. To retrieve information about a registered UE, to update information, or to deregister a UE, an application sends a request under http://\{apiRoot\}/rim/v1/userEquipments/\{ueID\}.

The ueFNs resource represents a list of all of the registered functional identities on an individual UE. An HTTP GET method retrieves the list of registered functional identities of given UE, and an HTTP POST method registers a new functional number for given registered UE. The resource is represented by the ueFNlist data structure, which is a structure of UE functional numbers and a self-referring URL.

The ueFNID resource refers to individual UE functional number information and it is represented by ueFNinfo, which is a structure of the UE functional number, UE functional number context status, and a self-referring URL. The allowed HTTP methods are GET, which retrieves information about an individual UE functional number, and DELETE is used to unregister a UE functional number.

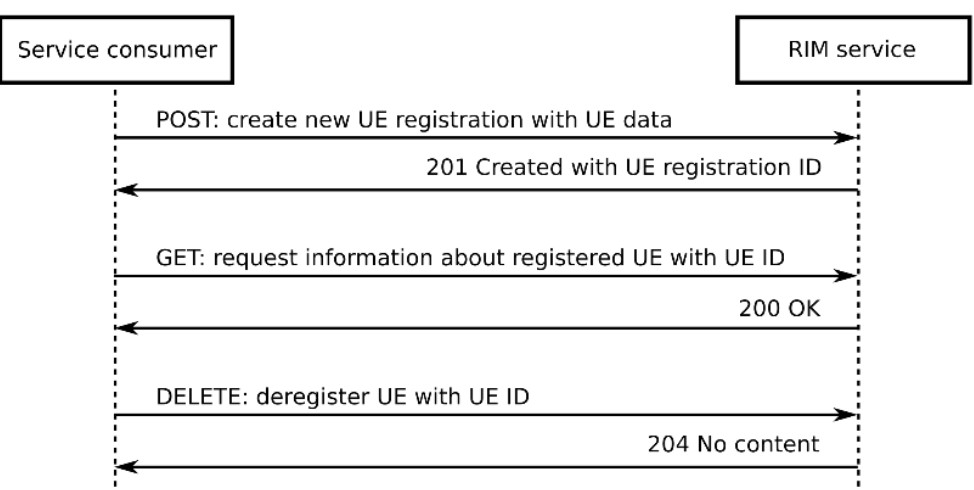

**Figure 6.** Flow for registering new UE, retrieving information about the UE, and deregistering the UE.

The users resource represents a list of all of the registered users on a given registered UE. An HTTP GET method retrieves the list of all of the registered users on an individual UE, and an HTTP POST method is used to register a new user on given registered UE, where the method body contains user credentials. The resource is represented by the userList data structure, which is a structure containing information about all users and a self-referring URL.

The userID resource refers to information about an individual user who is registered on a UE. The information is represented by a userInfo data structure that comprises information about user presence (context) status and a self-referring URL. The allowed HTTP methods are GET, which retrieves user information, and DELETE, which is used to unregister an individual user.

The userFIDs resource represents a list of all of the registered functional identities of a user. The HTTP GET method retrieves all of the registered user functional identities, and the HTTP POST method is used to register a new user functional identity. The resource is represented by the userFIDList data structure, which comprises a collection of user functional identity information (userFIDinfo) and a self-referring URL.

The userFID resource refers to information about an individually registered functional identity of a user. The information is represented by userFIDInfo, which comprises the functional identity information and a self-referring URL. The allowed HTTP methods are GET, which retrieves user functional identity information, and DELETE, which is used to unregister an individual user functional identity.

The ueSubscriptions resource represents all the subscriptions for notifications regarding the registration status of registered UEs. An application uses an HTTP GET method to retrieve a list of all the subscriptions and an HTTP POST method to create a new subscription. The ueSubscriptionInfo data type represents subscription data, and it includes: a correlator that is used by an application that tags the resource during a request to create a subscription; the address at which the application wants to receive notifications; the ueEventCriteria data type, which is a list of event values to generate notifications about the registration status of UEs; the subscription duration; and the self-referring URL. The ueNotification data type represents UE notifications, and it contains the UE address, the UE status related event type (registered, unregistered), timestamp, and a link to other resources related to this notification.

The ueSubscriptionID resource refers to an individual subscription and the allowed HTTP methods that may be used by an application are: GET, which retrieves information about an existing subscription; PUT, which updates an existing subscription; and DELETE, which terminates a subscription.

The ueFNSubscriptions resource, userSubscriptions resource, and userFIDSubscriptions resources represent all the subscriptions for notifications about registered UE functional numbers, users, and user functional identities, respectively. A GET method retrieves a list of the respective subscriptions, and a POST method creates a new subscription.

The ueFNSubscriptionID, userSubscriptionID, and userFIDSubscriptionID resources represent existing subscriptions for notifications about registered UE functional numbers, users, and user functional identities, respectively. An existing subscription is modified using an HTTP PUT method, it is terminated using an HTTP DELETE method, and read using an HTTP GET method.

When a UE is switched off, all the registered identities assigned to it are deregistered. The FRMCS system can also deregister any of the registered identities, and this event is reported to all applications with active subscriptions for notifications about changes in the identity registration status.

Figure 7 shows a scenario for controlling subscriptions to notifications about the registration status changes in UE functional numbers.

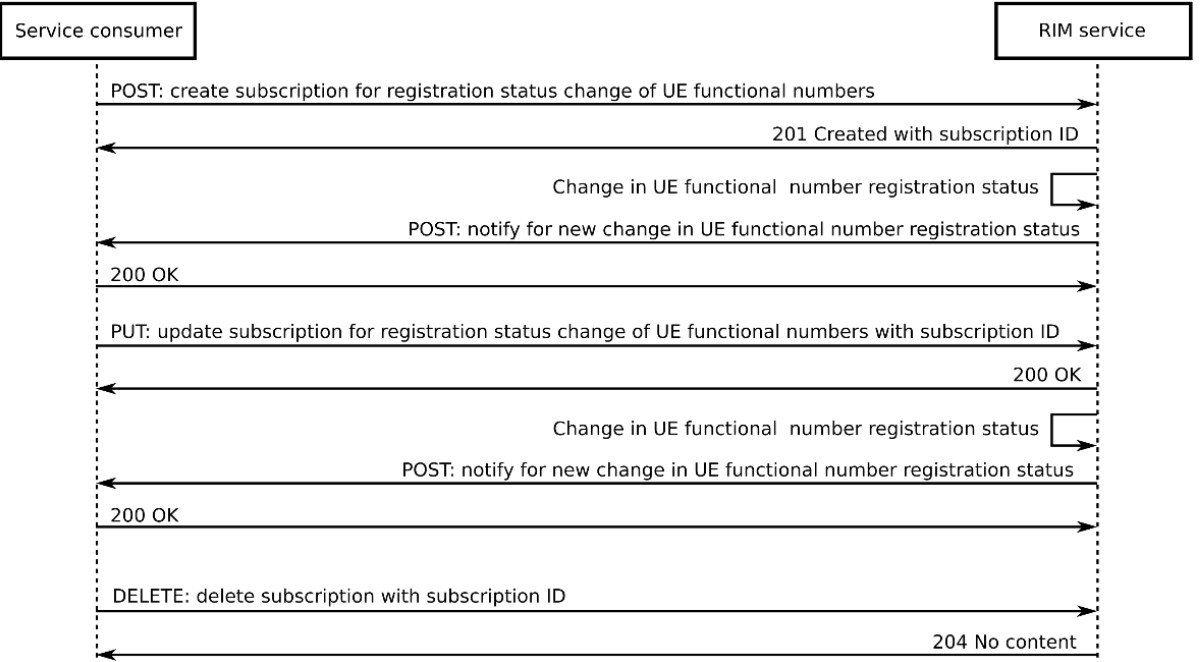

**Figure 7.** Subscription management and notifications about status changes in UE functional number registration.

To start a subscription, an application creates a new resource under http://\{apiRoot\}/rim/v1/ueSubscriptions/\{ueSubscriptionID\}/ueFNSubscriptions.

To update or terminate an existing subscription about changes in the registration status of a UE functional number, an application updates or deletes a resource under http://\{apiRoot\}/rim/v1/ueSubscriptions/\{ueSubscriptionID\}/ueFNSubscriptions/\{ueFNSubscriptionID}.

## 4. Modeling the Identity Registration Status

The applications, that use the RIM API, and the FRMCS system need to have synchronized views on the identity registration status.

Figure 8 shows the application's view on the registration status of "Cab radio" UE, its functional identities, user identity, and user functional identities.

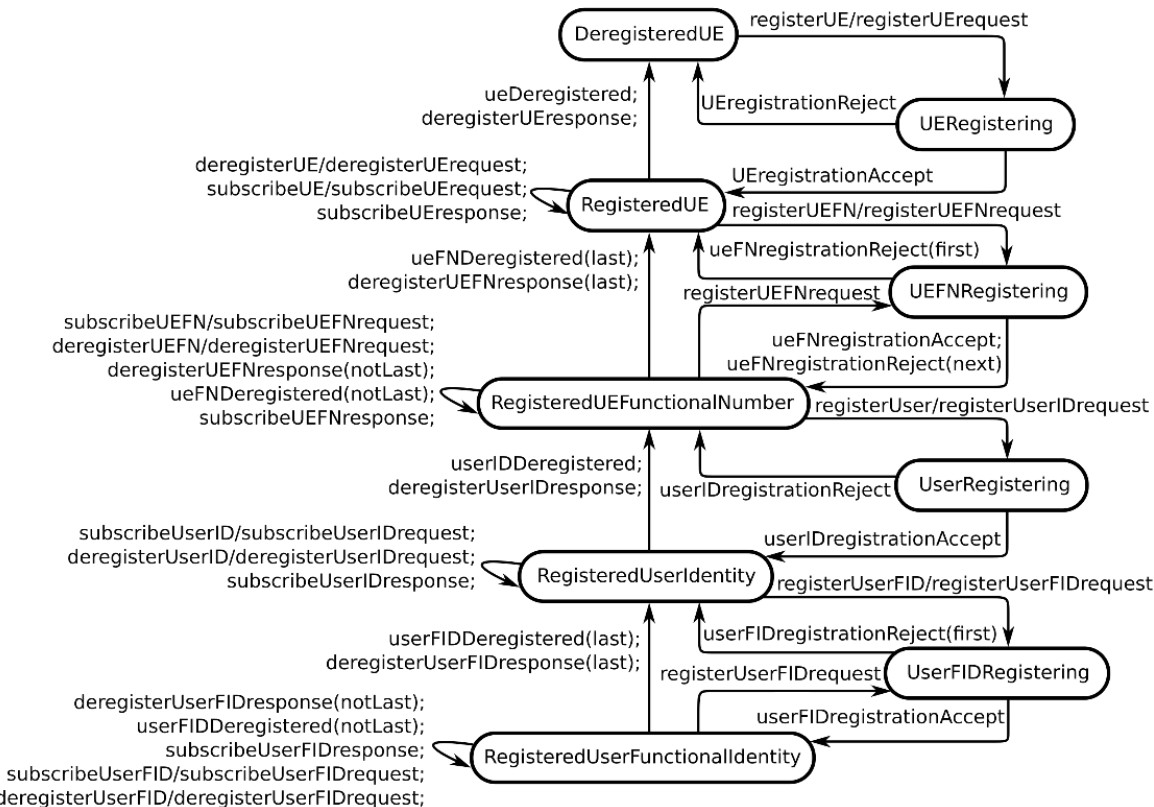

**Figure 8.** How the application views the registration status of "Cab radio" UE, its functional identities, user identity, and user functional identities.

In the DeregisteredUE state, the UE is attached to the FRMCS network, but it is not registered in the FRMCS system. FRMCS application(s) are ready for use. In the UERegistering state, the application waits for the approval/rejection of a UE identity from the FRMCS system. In the RegisteredUE state, the UE is logged in to the FRMCS system and the application may create a subscription for notifications about the UE registration status. In the UEFNRegistering state, the application waits for the approval/rejection of the UE functional numbers from the FRMCS system. In the RegisteredUEFunctionalNumber state, one or more UE functional number(s) are logged in to the FRMCS system, and the application may create a subscription for notifications about the registration status of UE functional numbers. In the UserRegistering state, the application waits for the approval/rejection of a user identity from the FRMCS system. In the RegisteredUserIentity state, the user is logged in, and the application may create a subscription for notifications about the user registration status. In the UserFIDRegistering state, the user waits for the approval/rejection of his/her functional identity from the FRMCS system. In the RegisteredUserFunctionalIdentity state, one or more user functional identities are registered into the FRMCS system, and the application may subscribe for notifications about the registration status of user functional identities. Both the user and FRMCS system may deregister any of the registered identities. Registrations may have a temporary duration, and, in this case, the FRMCS may request periodic registration status updates from the UE. If a registration update has not been received, the FRMCS system deregisters the respective identity.

Figure 8 is simplified, as it does not include the behavior in which a registration update is requested for brevity.

Figure 9 shows the FRMCS system's view on the registration status "Cab radio" UE, its functional identities, user identity, and user functional identities.

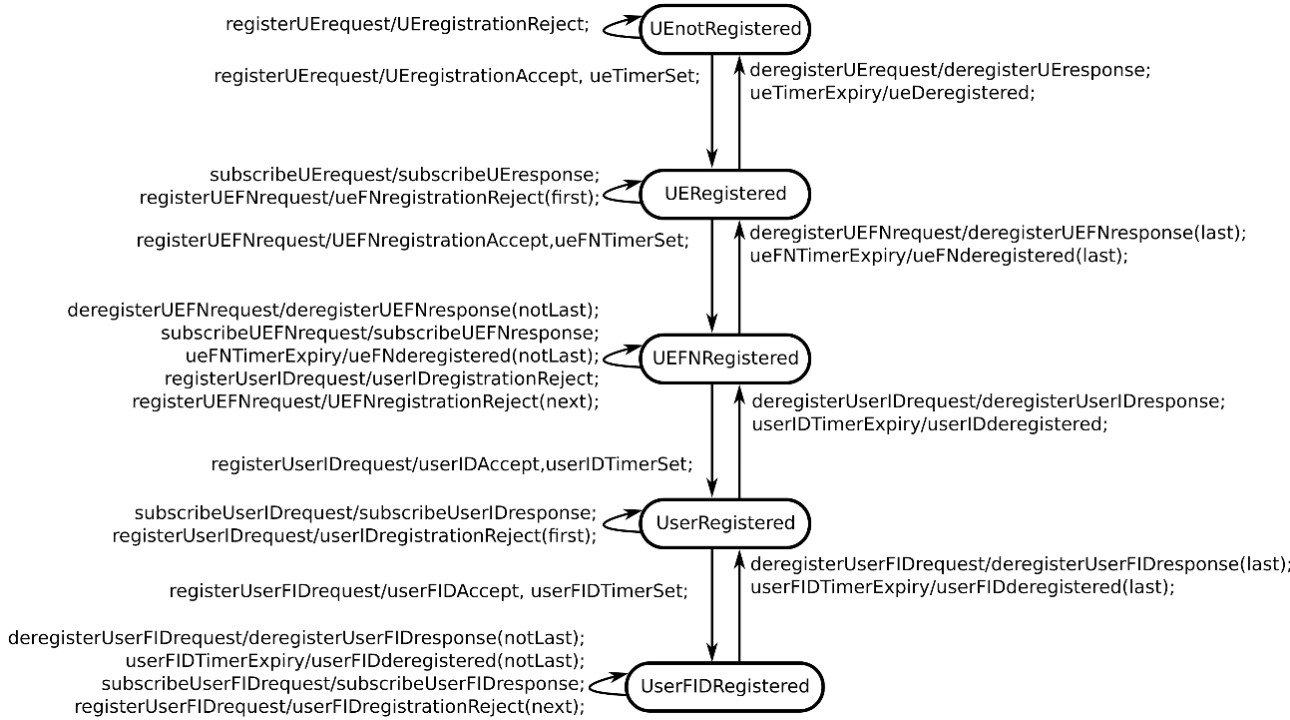

**Figure 9.** FRMCS system's view on the registration status of "Cab radio" UE, its functional identities, user identity, and user functional identities.

In the UEnotRegistered state, the UE is not registered in the FRMCS system. In the UERegistered state, the UE identity is registered in the FRMCS system. In the UEFNRegistered state, one or more UE functional numbers are logged in to the FRMCS system. In the UserRegistered state, the user is logged in to the FRMCS system. In the UserFIDRegistered state, one or more user functional identities are registered. In each registered state, the FRMCS system maintains the respective timers determining the valid registration duration. In case a request for a registration update has not been received, the FRMCS system deregisters the identity. Other events may also trigger the deregistration of any registered identity by the FRMCS system, e.g., due to administrative reasons (not shown in Figure 9). In cases where there are changes in the registration status, the FRMCS system notifies the applications with active subscriptions.

The models are simplified, as the figures do not show subscription termination in cases of identity deregistration.

To prove that the models expose equivalent behavior, i.e., they support synchronized views on the identity registration status, the models are formally described and a bisimulation relationship between their states is identified.

The formal model description uses a Labeled Transition System (LTS) to prove the behavioral synchronism of both state machines.

An LTS is a quadruple of a set of states, a set of labels (regarded as actions that drive the transitions), a set of transitions, and a set of initial states [20,21].

$L^{app} = (S^{app}, A^{app}, T^{app}, s_0^{app})$ denotes an LTS that describes the state machine representing the application's view of the registration status of different identities, where short names are in brackets, and:

$S^{app}$ = {DeregisteredUE [$s_1^a$], UERegistering [$s_2^a$], RegisteredUE [$s_3^a$], UEFNRegistering [$s_4^a$], RegisteredUEFunctionalNumber [$s_5^a$], UserRegistering [$s_6^a$], RegisteredUserIdentity [$s_7^a$], UserFIDRegistering [$s_8^a$], RegisteredUserFunctionalIdentity [$s_9^a$]},

$A^{app}$ = {registerUE [$i_1^a$], UEregistrationAccept [$i_2^a$], UEregistrationReject [$i_3^a$], subscribeUE [$i_4^a$], subscribeUEresponse [$i_5^a$], registerUEFN [$i_6^a$], ueFNregistrationAccept [$i_7^a$], ueFNregistrationReject(first) [$i_8^a$], ueFNregistrationReject(next) [$i_9^a$], subscribeUEFN

[$i_{10}{}^a$], subscribeUEFNresponse [$i_{11}{}^a$], registerUser [$i_{12}{}^a$], userIDregistrationAccept [$i_{13}{}^a$], userIDregistrationReject [$i_{14}{}^a$], subscribeUserID [$i_{15}{}^a$], subscribeUserIDresponse [$i_{16}{}^a$], registerUserFID [$i_{17}{}^a$], userFIDregistrationAccept [$i_{18}{}^a$], userFIDregistrationReject(first) [$i_{19}{}^a$], userFIDregistrationReject(next) [$i_{20}{}^a$], subscribeUserFID [$i_{21}{}^a$], subscribeUserFIDresponse [$i_{22}{}^a$], ueDeregistered [$i_{23}{}^a$], ueFNDeregistered (last) [$i_{24}{}^a$], ueFNDeregistered (notLast) [$i_{25}{}^a$], userIDDeregistered [$i_{26}{}^a$], userFIDDeregistered (last) [$i_{27}{}^a$], userFIDDeregistered (not-Last) [$i_{28}{}^a$], deregisterUE [$i_{29}{}^a$], deregisterUEresponse [$i_{30}{}^a$], deregisterUEFN [$i_{31}{}^a$], deregisterUEFNresponse(last) [$i_{32}{}^a$], deregisterUEFNresponse(notLast) [$i_{33}{}^a$], deregisterUserID [$i_{34}{}^a$], deregisterUserIDresponse [$i_{35}{}^a$], deregisterUserFID [$i_{36}{}^a$], deregisterUserFIDresponse(last) [$i_{37}{}^a$], deregisterUserFIDresponse(notLast) [$i_{38}{}^a$]},

$T^{app}$ = {($s_1{}^a$ $i_1{}^a$ $s_1{}^a$), ($s_2{}^a$ $i_2{}^a$ $s_3{}^a$), ($s_2{}^a$ $i_3{}^a$ $s_1{}^a$), ($s_3{}^a$ $i_4{}^a$ $s_3{}^a$), ($s_3{}^a$ $i_5{}^a$ $s_3{}^a$), ($s_3{}^a$ $i_6{}^a$ $s_4{}^a$), ($s_4{}^a$ $i_7{}^a$ $s_5{}^a$), ($s_4{}^a$ $i_8{}^a$ $s_3{}^a$), ($s_4{}^a$ $i_9{}^a$ $s_5{}^a$), ($s_5{}^a$ $i_{10}{}^a$ $s_5{}^a$), ($s_5{}^a$ $i_{11}{}^a$ $s_5{}^a$), ($s_5{}^a$ $i_{12}{}^a$ $s_6{}^a$), ($s_6{}^a$ $i_{13}{}^a$ $s_7{}^a$), ($s_6{}^a$ $i_{14}{}^a$ $s_5{}^a$), ($s_7{}^a$ $i_{15}{}^a$ $s_7{}^a$), ($s_7{}^a$ $i_{16}{}^a$ $s_7{}^a$), ($s_7{}^a$ $i_{17}{}^a$ $s_8{}^a$), ($s_8{}^a$ $i_{18}{}^a$ $s_9{}^a$), ($s_8{}^a$ $i_{19}{}^a$ $s_7{}^a$), ($s_8{}^a$ $i_{20}{}^a$ $s_9{}^a$), ($s_9{}^a$ $i_{21}{}^a$ $s_9{}^a$), ($s_9{}^a$ $i_{22}{}^a$ $s_9{}^a$), ($s_9{}^a$ $i_{28}{}^a$ $s_9{}^a$), ($s_9{}^a$ $i_{27}{}^a$ $s_7{}^a$), ($s_7{}^a$ $i_{26}{}^a$ $s_5{}^a$), ($s_5{}^a$ $i_{25}{}^a$ $s_5{}^a$), ($s_5{}^a$ $i_{24}{}^a$ $s_3{}^a$), ($s_3{}^a$ $i_{23}{}^a$ $s_1{}^a$), ($s_9{}^a$ $i_{36}{}^a$ $s_9{}^a$), ($s_9{}^a$ $i_{38}{}^a$ $s_9{}^a$), ($s_9{}^a$ $i_{37}{}^a$ $s_7{}^a$), ($s_7{}^a$ $i_{34}{}^a$ $s_7{}^a$), ($s_7{}^a$ $i_{35}{}^a$ $s_5{}^a$), ($s_5{}^a$ $i_{31}{}^a$ $s_5{}^a$), ($s_5{}^a$ $i_{33}{}^a$ $s_5{}^a$), ($s_5{}^a$ $i_{32}{}^a$ $s_3{}^a$), ($s_3{}^a$ $i_{29}{}^a$ $s_3{}^a$), ($s_3{}^a$ $i_{30}{}^a$ $s_1{}^a$) },

$s_0{}^{app}$ = {$s_1{}^{app}$}.

$L^{sys}$ = ($S^{sys}$, $A^{sys}$, $T^{sys}$, $s_0{}^{sys}$) denotes an LTS that describes the state machine representing the FRMCS system's view on the registration status of different identities, where

$S^{sys}$ = {UEnotRegistered [$s_1{}^s$], UERegistered [$s_2{}^s$], UEFNRegistered [$s_3{}^s$], UserRegistered [$s_4{}^s$], UserFIDRegistered [$s_5{}^s$]},

$A^{sys}$ = {registerUErequest [$i_1{}^s$], subscribeUErequest [$i_2{}^s$], registerUEFNrequest [$i_3{}^s$], subscribeUEFNrequest [$i_4{}^s$], registerUserIDrequest [$i_5{}^s$], subscribeUserIDrequest [$i_6{}^s$], registerUserFIDrequest [$i_7{}^s$], subscribeUserFIDrequest [$i_8{}^s$], ueTimerExpiry [$i_9{}^s$], ueFNTimerExpiry [$i_{10}{}^s$], userIDTimerExpiry [$i_{11}{}^s$], userFIDTimerExpiry [$i_{12}{}^s$], deregisterUErequest [$i_{13}{}^s$], deregisterUEFNrequest [$i_{14}{}^s$], deregisterUserIDrequest [$i_{15}{}^s$], deregisterUserFIDrequest [$i_{16}{}^s$]},

$T^{sys}$ = {($s_1{}^s$ $i_1{}^s$ $s_2{}^s$), ($s_1{}^s$ $i_1{}^s$ $s_1{}^s$), ($s_2{}^s$ $i_2{}^s$ $s_2{}^s$), ($s_2{}^s$ $i_3{}^s$ $s_3{}^s$), ($s_2{}^s$ $i_3{}^s$ $s_2{}^s$), ($s_3{}^s$ $i_4{}^s$ $s_3{}^s$), ($s_3{}^s$ $i_3{}^s$ $s_3{}^s$), ($s_3{}^s$ $i_5{}^s$ $s_4{}^s$), ($s_3{}^s$ $i_5{}^s$ $s_3{}^s$), ($s_4{}^s$ $i_6{}^s$ $s_4{}^s$), ($s_4{}^s$ $i_7{}^s$ $s_5{}^s$), ($s_4{}^s$ $i_7{}^s$ $s_4{}^s$), ($s_5{}^s$ $i_8{}^s$ $s_5{}^s$), ($s_5{}^s$ $i_7{}^s$ $s_5{}^s$), ($s_5{}^s$ $i_{12}{}^s$ $s_5{}^s$), ($s_5{}^s$ $i_{12}{}^s$ $s_4{}^s$), ($s_4{}^s$ $i_{11}{}^s$ $s_3{}^s$), ($s_3{}^s$ $i_{10}{}^s$ $s_2{}^s$), ($s_3{}^s$ $i_{10}{}^s$ $s_3{}^s$), ($s_5{}^s$ $i_{16}{}^s$ $s_4{}^s$), ($s_5{}^s$ $i_{16}{}^s$ $s_5{}^s$), ($s_4{}^s$ $i_{15}{}^s$ $s_3{}^s$), ($s_3{}^s$ $i_{14}{}^s$ $s_2{}^s$), ($s_3{}^s$ $i_{14}{}^s$ $s_3{}^s$), ($s_2{}^s$ $i_9{}^s$ $s_1{}^s$), ($s_2{}^s$ $i_{13}{}^s$ $s_1{}^s$)},

$s_0{}^{sys}$ = {$s_1{}^s$}.

The concept of bisimulation formalizes the idea of behavioral equivalence on different processes. Informally, two processes described as state machines are equivalent if each transition in one state machine corresponds to a transition in the other state machine and vice versa. This is the case when there is a strong bisimulation relationship. In a weak bisimulation, there may be internal transitions and states that are not observable. A formal definition of bisimulation may be found in [22,23].

**Proposition 1.** *$L^{app}$ and $L^{sys}$ are weakly bi-similar.*

**Proof.** Two LTSs have a weak bisimulation relationship if there is a peer–state relationship R between their states, such that for each transition from a state of the one LTS in R that terminates in another state of the same LTS in R, there is a corresponding transition from the peered state of the other LTS in R that terminates in the corresponding peered state.

Let R = {($s_1{}^a$, $s_1{}^s$), ($s_3{}^a$, $s_2{}^s$), ($s_5{}^a$, $s_3{}^s$), ($s_7{}^a$, $s_4{}^s$), ($s_9{}^a$, $s_5{}^s$)}. Then, the following transition mapping is identified:

1. The UE registration is successful, and the application subscribes for UE registration events: for $\forall$ ($s_1{}^a$ $i_1{}^a$ $s_2{}^a$) $\wedge$ ($s_2{}^a$ $i_2{}^a$ $s_3{}^a$) $\wedge$ ($s_3{}^a$ $i_4{}^a$ $s_3{}^a$) $\wedge$ ($s_3{}^a$ $i_5{}^a$ $s_3{}^a$) $\exists$ ($s_1{}^s$ $i_1{}^s$ $s_2{}^s$) $\wedge$ ($s_2{}^s$ $i_2{}^s$ $s_2{}^s$).

2. The UE registration fails: for $\forall$ ($s_1{}^a$ $i_1{}^a$ $s_2{}^a$) $\wedge$ ($s_2{}^a$ $i_3{}^a$ $s_1{}^a$) $\exists$ ($s_1{}^s$ $i_1{}^s$ $s_1{}^s$).

3. The first UE functional number is successfully registered, and the application subscribes for UE functional number registration events: for $\forall$ ($s_3{}^a$ $i_6{}^a$ $s_4{}^a$) $\wedge$ ($s_4{}^a$ $i_7{}^a$ $s_5{}^a$) $\wedge$ ($s_5{}^a$ $i_{10}{}^a$ $s_5{}^a$) $\wedge$ ($s_5{}^a$ $i_{11}{}^a$ $s_5{}^a$) $\exists$ ($s_2{}^s$ $i_3{}^s$ $s_3{}^s$) $\wedge$ ($s_3{}^s$ $i_4{}^s$ $s_3{}^s$).

4.　The first UE functional number registration fails: for $\forall$ ($s_3{}^a$ $i_6{}^a$ $s_4{}^a$) $\wedge$ ($s_4{}^a$ $i_8{}^a$ $s_3{}^a$) $\exists$ ($s_2{}^s$ $i_3{}^s$ $s_2{}^s$).

5.　The UE registers an additional functional number successfully, and the application subscribes for its status: for $\forall$ ($s_5{}^a$ $i_6{}^a$ $s_4{}^a$) $\wedge$ ($s_4{}^a$ $i_7{}^a$ $s_5{}^a$) $\exists$ ($s_3{}^s$ $i_3{}^s$ $s_3{}^s$) $\wedge$ ($s_3{}^s$ $i_4{}^s$ $s_3{}^s$).

6.　The UE's attempt to register an additional functional number fails: for $\forall$ ($s_5{}^a$ $i_6{}^a$ $s_4{}^a$) $\wedge$ ($s_4{}^a$ $i_9{}^a$ $s_5{}^a$) $\exists$ ($s_3{}^s$ $i_3{}^s$ $s_3{}^s$).

7.　The user is logged in to the FRMCS system successfully, and the application subscribes for user registration events: for $\forall$ ($s_5{}^a$ $i_{12}{}^a$ $s_6{}^a$) $\wedge$ ($s_6{}^a$ $i_{13}{}^a$ $s_7{}^a$) $\wedge$ ($s_7{}^a$ $i_{15}{}^a$ $s_7{}^a$) $\wedge$ ($s_7{}^a$ $i_{16}{}^a$ $s_7{}^a$) $\exists$ ($s_3{}^s$ $i_5{}^s$ $s_4{}^s$) $\wedge$ ($s_4{}^s$ $i_6{}^s$ $s_4{}^s$).

8.　The user login to the FRMCS system fails: for $\forall$ ($s_5{}^a$ $i_{12}{}^a$ $s_6{}^a$) $\wedge$ ($s_6{}^a$ $i_4{}^a$ $s_5{}^a$) $\exists$ ($s_3{}^s$ $i_5{}^s$ $s_3{}^s$).

9.　The registration of the first user functional identity is successful, and the application subscribes for user functional identity registration events: for $\forall$ ($s_7{}^a$ $i_{17}{}^a$ $s_8{}^a$) $\wedge$ ($s_8{}^a$ $i_{18}{}^a$ $s_9{}^a$) $\wedge$ ($s_9{}^a$ $i_{21}{}^a$ $s_9{}^a$) $\wedge$ ($s_9{}^a$ $i_{22}{}^a$ $s_9{}^a$) $\exists$ ($s_4{}^s$ $i_7{}^s$ $s_5{}^s$) $\wedge$ ($s_5{}^s$ $i_8{}^s$ $s_5{}^s$).

10.　The registration of the first user functional identity fails: for $\forall$ ($s_7{}^a$ $i_{17}{}^a$ $s_8{}^a$) $\wedge$ ($s_8{}^a$ $i_{19}{}^a$ $s_7{}^a$) $\exists$ ($s_4{}^s$ $i_7{}^s$ $s_4{}^s$).

11.　The user successfully registers an additional user functional identity, and the application subscribes for its registration status: for $\forall$ ($s_9{}^a$ $i_{17}{}^a$ $s_8{}^a$) $\wedge$ ($s_8{}^a$ $i_{18}{}^a$ $s_9{}^a$) $\exists$ ($s_5{}^s$ $i_7{}^s$ $s_5{}^s$) $\wedge$ ($s_5{}^s$ $i_8{}^s$ $s_5{}^s$).

12.　The user attempts to register another user functional identity and fails: for $\forall$ ($s_9{}^a$ $i_{17}{}^a$ $s_8{}^a$) $\wedge$ ($s_8{}^a$ $i_{20}{}^a$ $s_9{}^a$) $\exists$ ($s_5{}^s$ $i_7{}^s$ $s_5{}^s$).

13.　The application is notified that a user functional identity (not the last one) has been deregistered by the FRMCS: for $\forall$ ($s_9{}^a$ $i_{28}{}^a$ $s_9{}^a$) $\exists$ ($s_5{}^s$ $i_{12}{}^s$ $s_5{}^s$).

14.　The application is notified that the last user functional identity has been deregistered by the FRMCS: for $\forall$ ($s_9{}^a$ $i_{27}{}^a$ $s_7{}^a$) $\exists$ ($s_5{}^s$ $i_{12}{}^s$ $s_4{}^s$).

15.　The application is notified that the user has been deregistered by the FRMCS: for $\forall$ ($s_7{}^a$ $i_{26}{}^a$ $s_5{}^a$) $\exists$ ($s_4{}^s$ $i_{11}{}^s$ $s_3{}^s$).

16.　The application is notified that one of the UE functional numbers (not the last one) has been deregistered by the FRMCS system: for $\forall$ ($s_5{}^a$ $i_{25}{}^a$ $s_5{}^a$) $\exists$ ($s_3{}^s$ $i_{10}{}^s$ $s_3{}^s$).

17.　The application is notified that the last UE functional number has been deregistered by the FRMCS system: for $\forall$ ($s_5{}^a$ $i_{24}{}^a$ $s_3{}^a$) $\exists$ ($s_3{}^s$ $i_{10}{}^s$ $s_2{}^s$).

18.　The application is notified that the UE has been deregistered by the FRMCS system: for $\forall$ ($s_3{}^a$ $i_{23}{}^a$ $s_1{}^a$) $\exists$ ($s_3{}^s$ $i_9{}^s$ $s_1{}^s$).

19.　The user deregisters one of his/her functional identities (not the last one): for $\forall$ ($s_9{}^a$ $i_{36}{}^a$ $s_9{}^a$) $\wedge$ ($s_9{}^a$ $i_{38}{}^a$ $s_9{}^a$) $\exists$ ($s_5{}^s$ $i_{16}{}^s$ $s_5{}^s$).

20.　The user deregisters his/her last functional identity: for $\forall$ ($s_9{}^a$ $i_{36}{}^a$ $s_9{}^a$) $\wedge$ ($s_9{}^a$ $i_{37}{}^a$ $s_7{}^a$) $\exists$ ($s_5{}^s$ $i_{16}{}^s$ $s_4{}^s$).

21.　The user logs out of the FRMCS system: for $\forall$ ($s_7{}^a$ $i_{34}{}^a$ $s_7{}^a$) $\wedge$ ($s_7{}^a$ $i_{35}{}^a$ $s_5{}^a$) $\exists$ ($s_4{}^s$ $i_{15}{}^s$ $s_3{}^s$).

22.　One of the UE functional numbers (not the last one) is deregistered: for $\forall$ ($s_5{}^a$ $i_{31}{}^a$ $s_5{}^a$) $\wedge$ ($s_5{}^a$ $i_{33}{}^a$ $s_5{}^a$) $\exists$ ($s_3{}^s$ $i_{14}{}^s$ $s_3{}^s$).

23.　The last UE functional number is deregistered: for $\forall$ ($s_5{}^a$ $i_{31}{}^a$ $s_5{}^a$) $\wedge$ ($s_5{}^a$ $i_{32}{}^a$ $s_3{}^a$) $\exists$ ($s_3{}^s$ $i_{14}{}^s$ $s_2{}^s$).

24.　The UE is deregistered from the FRMCS system: For $\forall$ ($s_3{}^a$ $i_{29}{}^a$ $s_3{}^a$) $\wedge$ ($s_3{}^a$ $i_{30}{}^a$ $s_1{}^a$) $\exists$ ($s_2{}^s$ $i_{13}{}^s$ $s_1{}^s$).

Therefore, R is a weak bisimulation relationship, and $L^{app}$ and $L^{sys}$ are weakly bisimilar, i.e., they expose equivalent behavior. □

The description of the formal model and the concept of weak bisimulation are used to check the model and to verify the behavioral equivalence of the identity registration processes in an application, as well as in the FRMCS system.

## 5. Location Management of Registered Identities

FRMCS standards do not provide any details regarding identity registration and do not specify how identities are managed. In this paper, it is supposed that there is a central

register (CR) where all UEs, their functional identities, user identities, and user functional identities, which are administered by a railway operator, are enlisted statically. For each identity, the CR has to store the identity's serving profile, which describes the authorizations that each identity has and which groups with special privileges that it belongs to. For each identity, the CR also needs to store the identity of the FRMCS system in which the UE and user are registered. Because there are expected to be a great deal of "Sensor"-type UE installed at trackside and on the trains, as well as other types of UE, due to the number of trains and controllers, drivers, train staff members, trackside staff members, other railway staff members, and members of the shunting teams, it is not expected that the information of all of the registered identities will be stored in one FRMCS system. If all of the registered identities are stored in one FRMCS system, then the signaling load to this FRMCS system would be too large.

To reduce the signaling load, it is likely that several FRMCS systems will be distributed throughout the railway operator network. In such an arrangement, each FRMCS system is responsible for managing the identities that are registered in its serving area. When UE is switched on, it registers itself in the FRMCS system that it visits (visited FRMCS system). If registration is successful, then the visited FRMCS system stores the registered identities. Once registered, it is not expected that the UE or users will have to register again. As such, the registered identities (and their serving profiles) related to moving trains and users have to be transferred to the new (target) FRMCS system and deleted from the old one. The CR also needs to support information about all of the FRMCS systems in the operator's railway network.

The functionality of the location management of registered identities may be implemented as another RESTful service called an Identity Location Management (ILM) service.

The ILM service enables the service consumer to do the following:

- Retrieve the FRMCS system list and information for the individual FRMCS system;
- Retrieve the UE list and individual UE information, including UE functional identities;
- Retrieve the user list and individual user information, including his/her functional identities;
- Update information about the serving FRMCS system for UEs and users and their functional identities;
- Manage the subscriptions to notifications about the UE, user, and functional identity registrations in FRMCS-serving areas;
- Manage the subscriptions to notifications about changes in the registration status of UEs, users, and functional identities.

Figure 10 shows the structure of the ILM service resource URI, where the ILM service URI is registered in a service directory.

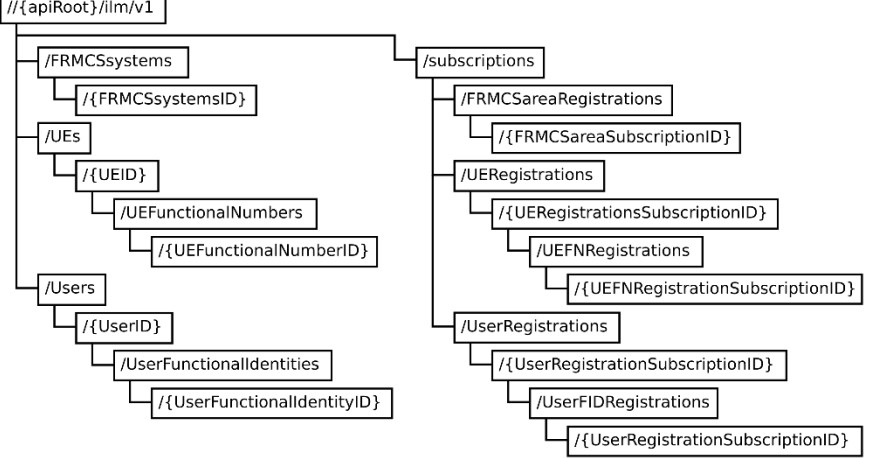

**Figure 10.** Structure of the ILM service resource URIs.

The FRMCSsystems resource is a container for all of the FRMCS systems defined in the operator's railway network. An HTTP GET method retrieves a list of all of the FRMCS systems.

The FRMCSsystemID resource represents individual FRMCS system information, including its identity and serving area. An HTTP GET method retrieves information about the individual FRMCS system.

The UEs resource represents the list of all of the UEs in the railway operator networks (both registered and unregistered). The HTTP GET method retrieves the list of UEs.

The UEID resource represents an individual UE. An HTTP GET method retrieves information about given UE, including information on the FRMCS system in which it is registered and its serving profile. An HTTP PATCH method is used to change the FRMCS system in which the UE is registered.

The UEFunctionalNumbers resource represents all of the functional numbers related to the UE, and an HTTP GET method is used to retrieve the list of UE functional numbers.

The UEFunctionalNumberID resource is for an individual UE functional number. Information about it is retrieved by an HTTP GET, and its serving FRMCS system is updated by an HTTP PATCH method.

The Users resource represents the list of all of the users in the railway network. An HTTP GET method retrieves the user list.

The UserID resource represents an individual user. Information about an individual user can be retrieved using an HTTP GET method. The FRMCS system in which the user is registered is updated by an HTTP PATCH method.

The UserFunctionalIdentities resource represents all of the functional identities of the user, and an HTTP GET method is used to retrieve a list of all of the user functional identities.

The UserFunctionalIdentityID resource is for an individual user functional identity. Information about the resource is retrieved by an HTTP GET method, while the serving FRMCS is updated using an HTTP PATCH method.

The subscriptions resource represents all of the subscriptions related to identity registration management.

The FRMCSareaRegistrations resource represents the subscriptions for notifications about registration changes in FRMCS-serving areas. An HTTP GET method returns all of the subscriptions of this type, and an HTTP POST method creates a new subscription.

The FRMCSareaSubscriptionID resource is for given subscription for registration change notifications and it allows an HTTP GET method to retrieve information about the subscription, an HTTP PUT method to update the subscription, and an HTTP DELETE method to terminate the subscription.

The UERegistrations resource, UEFNRegistrations resource, UserRegistrations resource, and UserFIDRegistrations resource represent all of the subscriptions to notifications about registration/deregistration events related to UE, UE functional numbers, users, and user functional identities, respectively. An HTTP GET method retrieves information about all the subscriptions, and an HTTP POST method creates a new subscription for each subscription type.

The UERegistrationSubscriptionID resource, UEFNRegistrationSubscriptionID resource, UserRegistrationSubscriptionID resource, and UserFIDRegistrationSubscriptionID resource represent individual subscriptions to notifications about registration/deregistration events related to UE, UE functional numbers, and users and user identities, respectively. The supported HTTP methods are GET, PUT, and DELETE, used to retrieve, update, and delete information about the subscription.

A UE and a user only need to register its and her/his identity and the related functional identities once. Events related to UE and user movements across the serving areas of different FRMCS systems can be reported by the FRMCS location service. When a UE enters a serving area of a new FRMCS system, the registration procedure for the UE and user identities in the new FRMCS system is triggered without the participation of the UE or the user.

Figure 11 shows the flow for the initial registration of a piece of UE and its functional numbers in FRMCS system 1 and subsequent registration in FRMCS system 2.

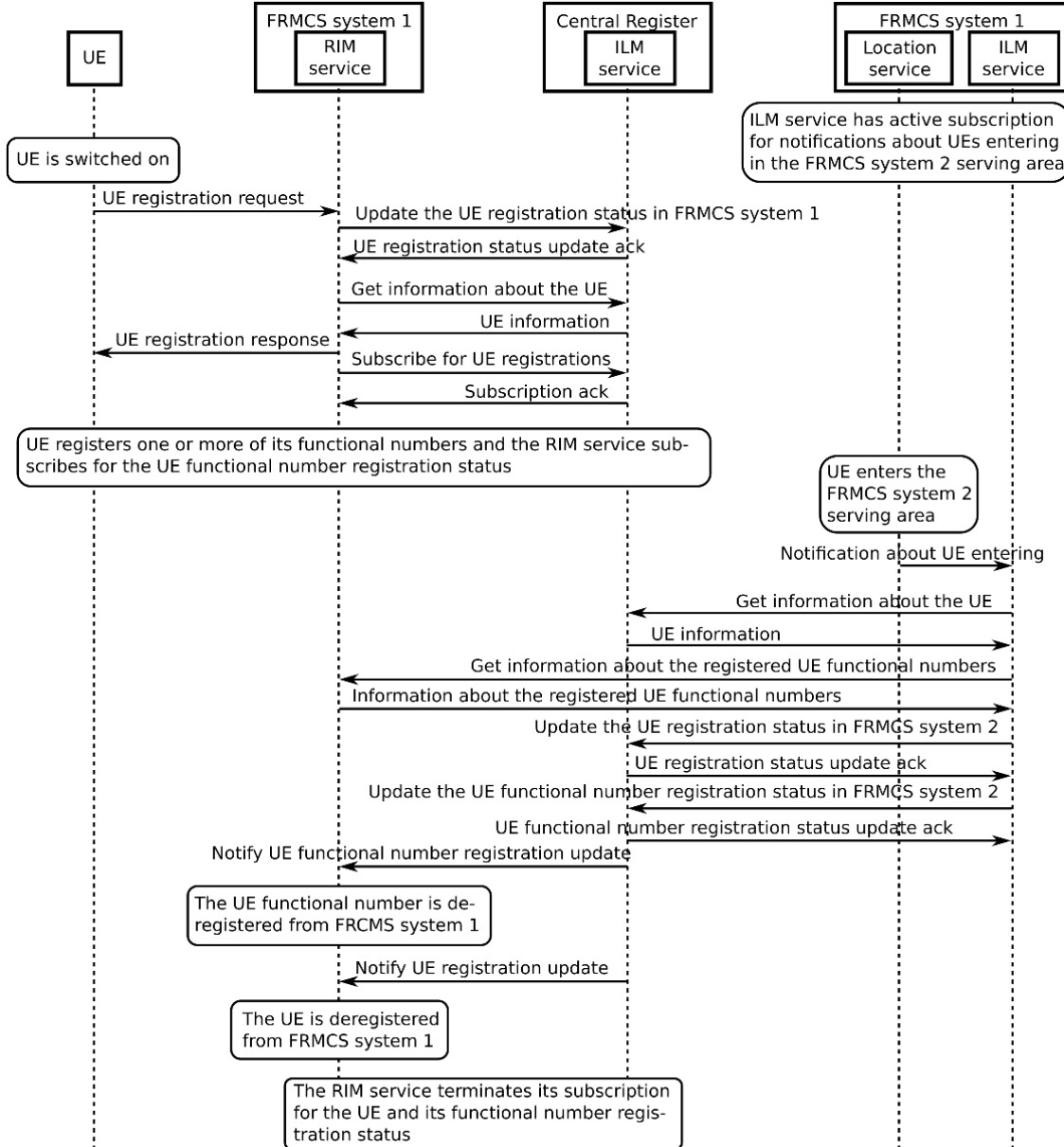

**Figure 11.** Flow of initial UE registration and subsequent registration in a new FRMCS system.

As a precondition, each FRMCS system knows which UE can enter and leave its serving area in advance. In the considered use case, the ILM service instance in FRMCS system 2 subscribes to notifications from the location service about the UE entering and leaving its serving area. The registration procedure starts by switching on the UE.

1. Once the UE is switched on, it sends a registration request to the RIM service in FRMCS system 1, as described in previous sections. The RIM service requests an update regarding the UE registration status using the ILM service in the CR by means of a PATCH request.
2. The CR acknowledges the update and stores the FRMCS system 1 identity as serving the UE.
3. The RIM service queries the CR for the UE serving profile and for its functional numbers if the UE is of the "Cab radio" or "PA" types by means of GET request.

4. The CR provides the requested information.
5. The RIM service acknowledges the UE registration.
6. The RIM service subscribes to notifications about UE registration status with the ILM service in the CR using a POST request.
7. The UE may register one or more of its functional numbers, and in the case of successful registration, the RIM service in the CR subscribes for their registration status.
8. While moving, the UE enters the serving area of FRMCS system 2.
9. The location service in FRMCS system 2 notifies the ILM service instance in FRMCS system 2 about the UE entering its serving area.
10. As a client of the ILM service, FRMCS system 2 retrieves information about the UE from the ILM service in the CR using a GET request.
11. The CR sends the requested information, including the identity of the FRMCS system where the UE is registered.
12. The ILM service in FRMCS system 2 queries the RIM service in FRMCS system 1 for information about UE serving profile and the registered UE functional numbers.
13. The RIM service provides information about the UE serving profile and registered functional numbers for different pieces of UE and their profiles.
14. Now that information about the registered UE and its registered functional numbers is available, and the ILM service client in FRMCS system 2 requests an update in the CR of the FRMSC serving area using a PATCH request.
15. The CR acknowledges the updated registration status of the UE.
16. The CR notifies the RIM service in FRMCS system 1 about the change in the registration status of UE and its functional numbers.
17. The RIM service terminates its subscription to the UE and to its functional numbers' registration status with the ILM service in the CR and considers the UE as being unregistered.

## 6. Identity Management in FRMCS Security and Safety

In [11], the required features and functions of an FRMCS system are defined. The main attributes of a secure FRMCS system include data integrity, data confidentiality, information privacy, the non-repudiation of data origin, and availability. Three independent security layers for FRMCS system are defined:

- In the railway application stratum, the security mechanisms are application-specific and are independent of the FRMCS.
- In the service stratum, the security mechanisms exist between FRMCS service clients and the FRMCS server, and include authentication, data integrity, data confidentiality, and data privacy.
- In the transport stratum, the security mechanisms are independent of the railway application stratum and service stratum.

The proposed services can be used to implement the required functionality of the security mechanisms in the service stratum.

In the service stratum, integrity can protect all types of data (e.g., user data and positioning information), and identity management may be used as a key integrity function. The confidentiality in the service stratum also aims to protect all types of data, and identity management may be used as a key function during the encryption process. The privacy in the service stratum protects the integrity and confidentiality of the equipment and user identities and may be implemented using one-time identities with limited validity. In the service stratum, non-repudiation allows the origin of data to be determined and includes functions for access protection, internal security, and interface security. Identity management and role management may be used during access protection for identification, authentication, and authorization. Internal security functions and the security functions of interfaces from/to external systems are implementation dependent.

The specifics of the security mechanism are not defined in the FRMCS standard and should be studied in future work.

The security of the communications between FRMCS service clients and FRMCS servers is an issue related to the transport stratum and may rely on the proven Internet Protocol Security (IPSec) architecture, Hypertext Transfer Protocol Secure (HTTPS) for RESTful services, and the elaborated security mechanisms of fifth-generation mobile networks.

Safety is a key requirement for railway operation because if it is not managed properly, then passengers, on-board and trackside staff, and infrastructure damages may occur. FRMCS standards define use cases for on-train safety devices to ground communications [10]. The use cases include:

- The initiation of on-board safety device data communication,
- The termination of on-board safety device data communication,
- Voice communication between the driver and controller.

These use cases utilize the identity management functionality. According to [10], a driver safety device is an on-board system that monitors driver alertness and issues warnings and alarms to other systems as appropriate. Pre-conditions assume that the identities of the user and equipment are registered and authorized to initiate ground data communication in the safety devices. Voice and/or data communication is automatically setup in case of emergency, e.g., by a driver safety device using an assured data communication service. Train information such as the functional identity is sent to the ground user. The functional identity provided by role management and presence applications as well as other information is used by the FRMCS system to determine the responsible controllers and the ground system in question. During the communication, the information gained from the role management and presence applications is used to present the identity of the on-train device to the controller.

## 7. Conclusions

Registering the identities of railway equipment, users, and functional roles enables identification, authentication, and authorization. The registered user identity, functional identity of a user, and functional identity of the equipment are presented in communication sessions. This paper presents a service-oriented approach for the management of equipment, users, and functional identities in FRMCS. Two services are proposed. Both services were designed according to the REST principles.

The Railway Identity Management service enables the initial registration of railway equipment, its functional numbers, users, and user functional identities, as well as identity deregistration by the FRMCS system or deregistration when the equipment is switched off or when the user logs out. The registration status models, supported by equipment applications and by FRMCS system, are designed, formally described, and theoretically verified.

The Identity Location Management service enables updates of the serving FRMCS system and transfer of registered identities and their serving profiles when the equipment or user roam between different FRMCS systems.

The main advantages of the proposed service-oriented approach for identity management are as follows:

- Virtualization of FRMCS functionality. The proposed services can run as virtual machines on the virtualized infrastructure deployed at the edge of the railway network. Network function virtualization enables robust software installation across railway network locations and eliminates the need for a specialized hardware infrastructure.
- Flexibility. Security services need to be more adaptive and have the benefits of easy provisioning and installation. The proposed service-oriented approach enables the seamless deployment of new enhanced security services.
- Reusability. Railway applications may be developed by accumulating self-contained, small, and functionally loosely coupled services. The services may be reused in multiple applications independently, as they provide basic capabilities.
- Location independence. The depicted services can be located through a service registry and can be accessed through their unique identifiers and can thus be relocated over time without interrupting service continuity.

- Reliability. The proposed small and independent services make the process of testing and debugging of applications easier.
- Scalability. The ever-increasing user demands for security require more responsible approaches to service provisioning. The proposed solution provides flexibility and agility to scale the security services across FRMCS servers.
- Cost saving. The proposed approach helps railway operators to achieve the greater efficiency of capital expenditures and reduces maintenance costs.

The cost/benefit rationale of the proposed service-oriented approach is at least twofold. The application of software on standard platforms that meet the demands of throughput and latency requires considerable processing resources. Services send and receive messages, and the flow intensity often reaches high levels, so high-bandwidth servers might be required. Further, the service-oriented approach introduces an extra payload and increased latency. The virtualization of FRMCS functions comes at the cost of more energy consumption.

The current level of privacy, i.e., maintaining the confidentiality and integrity of identities, is entirely reliant on the privacy provided by communication protocol stacks. Future work related to railway identity management may be oriented towards the provisioning of identity privacy. As identities are transferred via radio links in FRMCS system, they are susceptible to security threats. Independent from the applied security mechanisms in next-generation mobile networks, identity confidentiality might be protected at the application layer. One possible solution is to use one-time identities with limited validity. Instead of transferring the unique equipment and user identities over the radio link, these one-time identities can be used instead. A new one-time identity can be assigned any time the equipment or user is involved in communication. The application of one-time identities requires procedures to be developed for their management.

**Author Contributions:** Conceptualization, E.P. and V.T.; methodology, E.P. and I.A.; validation, I.A. All authors have read and agreed to the published version of the manuscript.

**Funding:** This research was funded by the Bulgarian National Science Fund under Grant No. KP-06-H57/12.

**Conflicts of Interest:** The authors declare no conflict of interest.

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
