# Peer review of "Identity Management in Future Railway Mobile Communication System"

_applsci, doi:10.3390/app12094293_

Round 1

Reviewer 1 Report

Dear Authors,

The paper's aim has been achieved systematically, and the paper is well organized. However, below are some points that need to consider:

1- The abstract needs to be improved as its current form does not adequately reflect what is conducted in the paper.

2- Many symbols and abbreviations are mentioned without a prior definition or defined many times, so please check.

Regards 

Author Response

The authors are grateful for valuable comments.

Comment 1: The abstract needs to be improved as its current form does not adequately reflect what is conducted in the paper.

            Answer: The abstract is rewritten and provides more details on the contribution.

Comment 2: Many symbols and abbreviations are mentioned without a prior definition or defined many times

            Answer: In the revised version, all abbreviations are described and duplications are removed.

Reviewer 2 Report

In this paper, the authors provides a Railway Identity Management(RIM) services and a Identity Location Management service to manage identity for Future Railway Mobile Communication System(FRMCS). This paper is hard to read. I have the following comments.

In section 1, the authors mentioned the importance of security and risk management for railway system. However, I can not find the safty test of the provided RIM service and Identity Location Management service. Maybe, I miss something.

In setion 2, the authors introduced the identities and registration management in FRMCS. The defination of user and user equipment(UE) and  functional identity is vague. They should be clearly explained. I am also confused about the registration process, what the difference of equipment type and the device?

Actually, the contents of this paper is so rich that the main problem and contribution of this paper merged together in the contents.  I suggest to reorgainze the sturcture of this paper to make it more clear and reasonable.

Author Response

The authors are grateful for valuable comments.

Comment 1: In section 1, the authors mentioned the importance of security and risk management for railway system. However, I cannot find the safety test of the provided RIM service and Identity Location Management service.

            Answer: In the revised version, a new section (Section 5 “Identity management in FRMCS security and safety”) is added which discusses the applicability of the proposed service-oriented approach to implementation of FRMCS security and safety features.

Comment 2: In section 2, the authors introduced the identities and registration management in FRMCS. The definition of user and user equipment (UE) and functional identity is vague. They should be clearly explained. I am also confused about the registration process, what the difference of equipment type and the device

            Answer: In the revised version, the definition of above mentioned concepts are presented as stated in the FRMCS standards with some additional clarifications added.

Comment 3: Actually, the contents of this paper is so rich that the main problem and contribution of this paper merged together in the contents. I suggest to reorganize the structure of this paper to make it more clear and reasonable.

            Answer: In the revised version, the “Introduction” is reworked and now it stresses more on existing works and highlights the contribution including the benefits of the proposed service-oriented approach. The last paragraph of the Introduction tries to reason about the paper structure providing more details on the content. Additionally, the body text of the conclusion is updated so to reflect the results in a clearer way while pointing the advantages and the limitations of the approach.

The revised paper has undergone recently a professional check done by the MDPI language editing service.

Reviewer 3 Report

This work presents a new identity management functionality for future railway mobile communication systems. Due to the interest of the topic addressed, I find the work of utility for the scientific community. However, I think that it could be suitable for publication in the Applied Sciences journal provided that the following comments are implemented within the document: 

- The authors should clearly state in the Introduction section the main advantages of the proposed identity management functionality as compared with previous related studies already published in the scientific literature, as well as the main problems overcome by the new proposal.
- Furthermore, the main limitations of the proposed solution should also be indicated.
- What mechanisms are foreseen for an eventual overflow produced by an excess of users?
- How robust is the functionality against potential security threats?
- What if the signal is suddenly interrupted? Is the system capable of continuing functionality when recovered?
- Some lines regarding future work to be performed by the authors in order to enhance the proposed solution should be added to the Conclusions section.
- The entire manuscript should be revised by a professional proofreading service.

Author Response

The authors are grateful for valuable comments.

Comment 1:

- The authors should clearly state in the Introduction section the main advantages of the proposed identity management functionality as compared with previous related studies already published in the scientific literature, as well as the main problems overcome by the new proposal.

Answer: In the revised version, the “Introduction” is revised and it stresses on existing works and highlights the contribution including the benefits of the proposed service-oriented approach. The main problem is also discussed in more details.

Comment 2: Furthermore, the main limitations of the proposed solution should also be indicated.

            Answer: In the revised version, the limitations of proposed service-oriented approach are discussed in the Conclusion.

Comment 3:

What mechanisms are foreseen for an eventual overflow produced by an excess of users?

            Answer: The paper studies only the identifications of the railway equipment and staff. As far as the proposed approach enables the virtualization of the FRMCS service, so two separate i. e. isolated network slices might be defined – one dedicated to the railway operation needs and another slice for the passengers’ needs. Thus, the impact of dynamic change in the number of passengers will not influence the operations’ slice, yet this is matter of configuration choice. However, when dealing with excessive loads, it is preferable to deploy new virtualized service instance(s), if possible at all, instead of going to the last line of defense like making request drops.

Comment 4:

How robust is the functionality against potential security threats?

            Answer: In the revised version, a new section (Section 5 “Identity management in FRMCS security and safety”) is added which discusses the applicability of the proposed service-oriented approach to implementation of FRMCS security and safety features.

Comment 5:

What if the signal is suddenly interrupted? Is the system capable of continuing functionality when recovered?

            Answer: The signal loss is a very fundamental problem, and the question is very good with respect to every radio technology. 5G New Radio copes with short term signal drops using different methods e. g. error correction through Automatic Repeat reQuest (ARQ), adopted from LTE. The longer signal drops problem is mitigated by the use of multiconnectivity which enables concurrent use of multiple communication paths. All reliable transport functions are key features of FRMCS transport stratum recovery, while the services we propose are in the FRMCS service stratum where the registers of identities and locations are storage functions.

Comment 6:

Some lines regarding future work to be performed by the authors in order to enhance the proposed solution should be added to the Conclusions section.

            Answer: In the corrected version, directions of future work are highlighted.

Comment 7:

The entire manuscript should be revised by a professional proofreading service.

            Answer: The revised paper has undergone recently a professional check done by the MDPI language editing service.

Round 2

Reviewer 2 Report

It is pleasing to see this revised edition. All of the proposed comments are well done. This paper is well organized, and the proposed issues are clearly explained. So, from my perspective, it is suitable for publication.